# Role of ERLINs in the Control of Cell Fate through Lipid Rafts

**DOI:** 10.3390/cells10092408

**Published:** 2021-09-13

**Authors:** Valeria Manganelli, Agostina Longo, Vincenzo Mattei, Serena Recalchi, Gloria Riitano, Daniela Caissutti, Antonella Capozzi, Maurizio Sorice, Roberta Misasi, Tina Garofalo

**Affiliations:** 1Department of Experimental Medicine, “Sapienza” University of Rome, 00161 Rome, Italy; valeria.manganelli@uniroma1.it (V.M.); agostina.longo@uniroma1.it (A.L.); serena.recalchi@uniroma1.it (S.R.); gloria.riitano@uniroma1.it (G.R.); danielacaissutti@gmail.com (D.C.); antonella.capozzi@uniroma1.it (A.C.); maurizio.sorice@uniroma1.it (M.S.); roberta.misasi@uniroma1.it (R.M.); 2Biomedicine and Advanced Technologies Rieti Center, “Sabina Universitas”, 02100 Rieti, Italy; vincenzo.mattei@uniroma1.it

**Keywords:** ERLINs, lipid rafts, MAMs, autophagy, apoptosis

## Abstract

ER lipid raft-associated protein 1 (ERLIN1) and 2 (ERLIN2) are 40 kDa transmembrane glycoproteins belonging to the family of prohibitins, containing a PHB domain. They are generally localized in the endoplasmic reticulum (ER), where ERLIN1 forms a heteroligomeric complex with its closely related ERLIN2. Well-defined functions of ERLINS are promotion of ER-associated protein degradation, mediation of inositol 1,4,5-trisphosphate (IP_3_) receptors, processing and regulation of lipid metabolism. Until now, ERLINs have been exclusively considered protein markers of ER lipid raft-like microdomains. However, under pathophysiological conditions, they have been described within mitochondria-associated endoplasmic reticulum membranes (MAMs), tethering sites between ER and mitochondria, characterized by the presence of specialized raft-like subdomains enriched in cholesterol and gangliosides, which play a key role in the membrane scrambling and function. In this context, it is emerging that ER lipid raft-like microdomains proteins, i.e., ERLINs, may drive mitochondria-ER crosstalk under both physiological and pathological conditions by association with MAMs, regulating the two main processes underlined, survival and death. In this review, we describe the role of ERLINs in determining cell fate by controlling the “interchange” between apoptosis and autophagy pathways, considering that their alteration has a significant impact on the pathogenesis of several human diseases.

## 1. Introduction

Preferential interactions between lipids and proteins lead to the organization of specialized sphingolipid-based microdomains on both plasma membrane and distinct organelle membranes in different cell types.

Indeed, evidence which has accumulated over the last two decades strongly supports the view that interactions between specific lipids, including cholesterol and sphingomyelin, leading to the formation of functionally important and relatively liquid-ordered (*L_o_*) domains, termed lipid rafts, which move within a fluid bilayer of cellular membranes, allowing the recruitment of other lipids and proteins [1,2]. Although cholesterol and sphingolipids are important rafts components, also gangliosides, sialic-acid containing glycosphingolipids (GSLs), are highly enriched in these regions where interact with cholesterol [3]. Consistent with these data, GSLs have been proposed as a core component of lipid rafts and are therefore used as typical lipid raft markers [4,5,6].

Development of methodologies available for their investigation clarified that these domains are distributed not only in both the outer and the inner leaflets of an asymmetric cell membrane, but also coupled across leaflets [7,8] to form functional domains with distinct compositions and properties [9,10]. Moreover, GSLs are asymmetrically enriched in the outer leaflet of the plasma membrane, suggesting that clustering of GSLs could be further stabilized by the formation of lateral carbohydrate-carbohydrate interactions [11].

Thus, a major feature of raft domains is to segregate specific elements with the aim to regulate their interactions with other membrane components, i.e., lipids and proteins, and hence their activity [2,12,13]. As multimolecular platforms, lipid rafts perform important functions through carbohydrate-interactions with specific proteins characterized especially by saturated lipid anchors GPI-like or palmitoyl moieties [14,15]. Some proteins are only transiently confined within these domains, while others are completely excluded [16], establishing the concept of transient and dynamic structures [17]. Recently, it has become evident that a complex network of lipid–lipid and lipid–protein interactions contribute to the activation of a variety of signaling pathways able to influence cell homeostasis [2]. In this way, lipid rafts contribute to regulate a variety of signal transduction pathways responsible for specific cellular programs, including apoptosis, proliferation, differentiation, stress responses, necrosis, inflammation, autophagy and senescence, thus determining cell fate [18].

## 2. Raft-like Microdomains in Internal Membranes

Lipid rafts, more specifically known as lipid “*raft-like microdomains*”, are distributed on the membrane of subcellular organelles, including ER, Golgi apparatus, endosomes, lysosomes and lipid droplets, as recently summarized in the review by Wang et al. [19], but also mitochondria [20,21,22], and nuclei [23]. At these sites, key reactions can be catalyzed with a significant impact in the regulation of intracellular trafficking and sorting [24], cholesterol homeostasis [25], and cell fate, i.e., survival or death [26,27,28,29]. In this way, they contribute to diverse biological processes [30].

Depending on the specific organelle, raft-like microdomains are particularly enriched in specific proteins, in close association with a certain lipid assortment, including cholesterol, glycosphingolipids and cardiolipin [31,32]. Although the key role of these structures in signal transduction is depending on protein composition, the characterization of lipid molecules as key components within raft-like microdomains has gained special attention in recent years [33,34,35]. Therefore, we focused on specific lipids associated with raft-like microdomains of principal organelles, which mainly contribute to cell fate.

### 2.1. Rafts-like Microdomains in the Mitochondria

It is well known that, although the 90% of total cellular cholesterol content is located in plasma membranes, a relatively low content of this lipid (around 3%) is also present in internal membranes [36]; nevertheless, as well as for plasma membranes, in internal membranes cholesterol is considered the major lipid component of “raft-like” microdomains, responsible for stabilizing protein and lipid interactions, leading to the formation of dynamic lipid platforms for internal signal transduction. Moreover, GD3 ganglioside, which is normally confined to plasma membrane lipid rafts, can be redistributed to mitochondrial membranes by actin cytoskeleton vesicular trafficking [37,38] in response to death signals.

A well described example of this statement is the redistribution of GD3 in lymphoblastoid T cells upon pro-apoptotic triggering induced by CD95/Fas ligation from plasma membrane (and/or from trans Golgi network) to the mitochondria lipid microdomains [38,39,40,41]. In this way, GD3 could promote both morphogenetic changes of mitochondrial membrane (i.e., curvature and membrane viscosity), and lead to the formation of clusters of apoptotic signaling molecules (t-Bid and Bax), which represent key events for apoptosis execution [42,43]. Moreover, disruption of lipid microdomains in isolated mitochondria using cholesterol sequestering agents selectively rescues the mitochondria depolarization induced by GD3 with consequent impairment of apoptosis [20]. In addition, molecules involved in the fission processes are associate with these microdomains. Indeed, hFis is constitutively present in mitochondrial raft-like microdomains, whereas dynamin-like protein 1 is recruited following proapoptotic stimulus. Thus, recruitment of fission-associated molecules to raft-like microdomains play a role in the morphogenetic changes leading to organelle fission [39]. On the other hand, localization of MFN2 in lipid rafts, via its molecular interaction with the ganglioside GD3, is mandatory in mitochondria network organization, playing a role in mitochondria fusion [41].

Moreover, cardiolipin, a mitochondrial phospholipid, was found as a crucial component within raft-like microdomains, specifically located at the contact sites formed between the inner and outer membranes [32], where acts as an activation platform for both caspase-8 recruitment, either contributing to regulate apoptosis, and for the recruitment of the multimolecular complex AMBRA1/BECN1/WIPI1 to regulate autophagy execution [44,45,46].

Thus, the inclusion of proteins into lipid rafts is tightly dependent on the lipid composition which becomes responsible for these preferential membrane sites, where membrane receptors are in close contact with target signaling molecules inducing the activation of signaling pathways of survival or death.

### 2.2. Rafts-like Microdomains in the Nuclei

For many years research focused on glycosphingolipids as lipid components of the inner nuclear membrane that influences the formation and stability of nuclear “rafts”. These structures represent essential platforms which strongly participate in maintaining the internal nuclear organization and function, influencing specific nuclear functions, including proliferation, differentiation, and apoptosis [47,48]. Lipid analysis of microdomains isolated from highly purified hepatocyte nuclei revealed a peculiar lipid composition characterized by high levels of phosphatidylcholine and sphingomyelin [23], partially linked with cholesterol. Cholesterol is considered essential to ensure lipid rafts formation on nuclear membrane, where it can exist in two principal pools: as sphingomyelin-free cholesterol without variation during cell proliferation and as sphingomyelin-linked cholesterol, which can be altered during the S-phase of the cell cycle when the nuclear-sphingomyelinase is activated [49].

These microdomains were proposed to act as platforms for transcription processes, as demonstrated by labeled (H^3^)-uridine incorporation during the S phase of the cell cycle [23]. Modification of nuclear microdomains induces a destruction of internal nuclear architecture with impairment of RNA transcription, suggesting a role for raft associated sphingomyelin in maintaining the nuclear integrity and function [50]. Moreover, exogenous cholesterol is also required for lipid accumulation and stabilization during cytokinesis [51,52].

Several lines of investigation indicate that control of cholesterol and sphingolipid metabolism is essential for regulation of signaling molecules associated to lipid rafts in mediating biological functions, such as cell survival and death [53]. In particular, studies in the recent past investigated the role of cholesterol in modulation of cell growth/proliferation and apoptosis [54]. Indeed, altered cholesterol metabolism occurs in a variety of cancers and contributes to tumor cell growth, since molecules derived from cholesterol, including steroid hormones, oxysterols and vitamin D can act as a ligand for estrogen-related receptor [55], with different and even opposite actions on cancer cells and tumor progression [56]. Depleting cholesterol from lipid rafts results in the disorganization of signal molecules and therefore increases the sensitivity of cancer cells to chemotherapy. Moreover, cholesterol depletion from lipid rafts in ovarian cancer cells, after increased cholesterol efflux due to specific transporters, is responsible for the phenotypic reprogramming of macrophages into tumor-associated macrophages, making them more responsive to pro-tumor signals, such as IL-4, and more resistant to the action of anti-tumor cytokines, such as interferon-gamma [53,54,55,56,57].

### 2.3. Rafts-like Microdomains in Golgi and ER

Endoplasmic reticulum (ER) stores only ~0.5–1% of the cell total cholesterol [58], comprising ∼3–5% of all ER lipids. However, functional raft-like microdomains have been described in the ER and in the Golgi apparatus under certain conditions [19].

In particular, the P24-P23 protein complex, which can bind to Sec24D, acting as a cargo receptor to mediate the export of GPI-anchored proteins like CD59 or the folate receptor may be associated with raft-like microdomains on the ER membrane. Cholesterol depletion disrupts the interaction between CD59 and P24-P23 and thus reduces the export of CD59 to Golgi, indicating that functional lipid rafts at the ER membrane can assist GPI-anchored proteins in transporting to Golgi [19].

In general, the subcellular distribution of lipid raft on internal membranes, including the Golgi apparatus or the ER, has a significant impact in the sorting of proteins and in the trafficking and overall exocytosis of viral proteins, which constitute fundamental steps in viral infection [59].

### 2.4. MAMs Raft-like Microdomains

Moreover, organelles communications have been reported to occur between ER and other organelles, such as mitochondria, through close physical contacts which are strongly modulated by lipid raft-like components. In this regard, emerging data support the existence of mitochondria-associated ER membranes (MAMs), which represent tethering sites between the membrane of ER and mitochondrial with 10–25 nm between them [60].

MAMs do not simply structurally link ER and mitochondria. An emerging role of MAMs is their ability in many signaling regulations, starting from the first discovery on the exchange of phospholipids [61], up to the control of metabolism and trafficking of different classes of lipids (i.e., cholesterol, sphingolipids, ceramide) [62] and proteins [63]. Thus, it is not surprising that the interplay between ER and mitochondria contributes during many circumstances to choose whether the cell should live or die.

Of interest, the biochemical analysis of purified MAM fractions revealed that they are characterized by the presence of specialized raft-like sub-domains enriched in cholesterol, which makes these membranes portions quite different from the remaining ones, allowing the membrane scrambling and contributing to the multiple functions of ER and mitochondria, respectively [64].

In this regard, since the major components of lipid microdomains reside within MAM subdomains [31,65], the role of gangliosides in regulating and influencing cellular activities through these subdomains has been investigated. In particular, gangliosides effect on cell fate could depend on structural characteristics and sugar modifications, as well as on their concentration [34,65]. GM1-ganglioside accumulation at MAMs can influence the activity of the inositol 1,4,5-trisphosphate (IP_3_) receptor (IP_3_R) by directly interacting with the channel. Consequently, this binding promotes an enhancement of Ca^2+^ transmission from the ER to the mitochondria, activating the mitochondrial apoptotic cascade [66].

Under stress conditions, cells can coordinate ER and mitochondrial functions to restore cellular homeostasis, involving MAMs raft-like microdomains, which contain many proteins that physically are able to make molecular bridges that regulate the close contact between the two organelles and that play important roles in lipid synthesis and Ca^2+^ transfer from the ER to mitochondria [67]. Both the mitochondria-shaping protein dynamin-related protein 1 (Drp1) [68] and Phosphofurin Acidic Cluster Sorting Protein 2 (PACS-2) [69], can have a function in regulating contacts between ER and mitochondria and both proteins were identified in lipid rafts. PACS-2, at ER level, regulates juxtaposition of the two compartments through BAP31-dependent fission and perinuclear clustering of mitochondria [70]. In the same way, Drp1 could alter tethering by causing fragmentation of mitochondria.

Proteomic analysis of raft-like microdomains within MAM revealed that the majority of the identified proteins are *bona fide* mitochondrial or ER proteins, according to the Gene Ontology annotation most of which have previously been noted as MAM-resident or -associated proteins. Furthermore, about 20% of the identified proteins have a documented association with lipid rafts. Most importantly, known internal lipid raft marker proteins (inositol 1,4,5-trisphosphate receptor type 3), ERLIN2, and voltage-dependent anion channel 1 (VDAC1) were detected in these domains, as well as most of the components of the mitochondrial/MAM-localized Ca^2+^ signaling complex [71]. Moreover, recently, Manganelli and colleagues defined a specific inter-organelle localization of ERLIN1 at MAM level [72].

## 3. ERLINs: Localization and Function

ER lipid raft-associated protein 1 (ERLIN1) is a transmembrane glycoprotein that forms a heteroligomeric complex with its closely related ERLIN2 towards the ER lumen. ERLIN1 and ERLIN2 belong to the prohibitin family of proteins since they contain a PHB domain. For this reason, they share many similar characteristics, including localization in cellular membranes, detergent insolubility, association with detergent resistant membranes and a propensity to form homo- and hetero-oligomers [73,74,75,76,77,78,79]. Furthermore, ERLIN1 and ERLIN2 are also known members of the Stomatin-prohibitin-flotillin-HflC/K (SPFH) domain–containing protein family, which includes stomatins, prohibitins, and flotillins [80]. In particular, members of the SPFH domain-containing proteins are associated to membranes of different intracellular compartments, including mitochondria (prohibitin), trans-Golgi network (flotillins), endosomes and plasma membrane (flotillin and stomatin) [79,81]. The association of these proteins with lipid raft–like microdomains [79] has led to the speculation that these proteins could bind specific classes of lipids. Several studies show, for example, that stomatin binds cholesterol [82], stomatin-like protein 2 binds cardiolipin [83], and prohibitin links to phosphatidylinositol (3,4,5)-trisphosphate PIP_3_ [84]. In fact, ERLINs not only bind cholesterol [85], but also associate to phosphatidylinositol 3-phosphate PI3P. ERLINs would provide a protein scaffold for the formation of specialized raft-like microdomains in the ER membrane, where they create a lipid microenvironment distinct from the rest of the ER membrane. Flotillins have been proposed to form “scaffolding microdomains” at the plasma membrane, which could provide platforms to include certain plasma membrane receptor signaling pathways [86] and a novel type of endocytosis [87]. Similarly, the “ERLIN microdomains” in the ER membrane could facilitate certain ER-associated processes, for example by clustering the proteins involved.

Since ERLIN1/2 complex seems to be localized exclusively at the ER [79,85,88] and ERLIN1/2 complex binds specifically to PI3P, it appears likely that the ERLIN1/2 complex–PI3P interaction may play a role in the ER functions.

By far, a well-defined function of ERLINS is to mediate IP_3_Rs processing [89] and regulation of lipid metabolism [85,90]. ERLINs promote ER-associated protein degradation (ERAD) of the activated IP3 receptor [88,91] and of 3-hydroxy-3-methylglutaryl- CoA reductase (HMGR) [92]. ER-associated protein degradation of several protein substrates has been shown to require ERLIN2 [91,93].

It has been shown that ERLIN2 is the dominant partner in the ERLIN1/2 complex and contains the determinants for binding to activated IP_3_Rs and PI3P. Interestingly, interaction of ERLIN2 with PI3P may be regulated by the Thr-65 region since the T65I mutation inhibits PI3P binding. Thus, some determinants of PI3P and activated IP_3_R are enriched in the same region of ERLIN2. It suggests, that PI3P may be a cofactor able to link activated IP_3_Rs to the ERLIN1/2 complex contributing to the retro-translocation of ubiquitinated IP_3_Rs from ER membrane [89].

Furthermore, several works have identified ERLINs as novel ER regulators of sterol regulatory element-binding proteins (SREBPs) that are crucial for cholesterol homeostasis. Cellular cholesterol levels are regulated by endoplasmic reticulum (ER) sterol sensing proteins, which include SREBP cleavage-activating protein (Scap) and Insulin-induced gene 1 (Insig1). SREBPs are transcription factors that dimerize with Scap in the event of low cellular cholesterol level. Under conditions of cholesterol sufficiency, cholesterol-bound Scap associates with Insig, which promotes ER retention of the SREBP–Scap complex [94]. However, when ER cholesterol decreases below a critical value, Scap undergoes a conformational change that allows packaging of SREBP–Scap in COPII-coated vesicles for subsequent transport to the Golgi. In the Golgi, site-specific proteases release the cytosolic transcription factor domain of SREBPs, that activates genes for cholesterol and fatty acid biosynthesis [94]. When cholesterol levels are restored, the SREBP–Scap–Insig complex accumulates in the ER [95]. In this scenario, ERLIN1 and ERLIN2 may suppress cholesterol production by blocking the export of (SREBPs) from the ER to the Golgi under high cholesterol conditions [85]. Because ERLINs bind to cholesterol and physically interact with SREBP–Scap–Insig, they could directly promote ER retention of SREBP–Scap. Thus, ERLINs could promote stability of the SREBP–Scap–Insig complex and may contribute to the highly cooperative control of the SREBP machine [85]. Alternatively, cholesterol association with ERLINs might nucleate the formation of cholesterol-rich microdomains in the ER that increase the interaction of Insig with SREBP–Scap. A schematic drawing of ERLINs as novel regulators of SREBP machinery is shown in Figure 1**.**

These data induce a reflection on the role that ERLIN could play as regards the structure and function of the lipid rafts, which may represent essential signaling platforms in the life-death balance of the cell [26,96].

Although ERLIN1 and ERLIN2 are well known as exclusively lipid raft-located proteins on ER membrane, little is known about their association with ER-MAM interface [97] and their involvement in autophagic initiation. In this concern, ERLIN contribution to the early phases of autophagosome formation was recently shown, suggesting that the interaction of ERLIN1 with autophagic proteins at lipid rafts is essential to promoting autophagy [72].

## 4. The Role of ERLINs in the Autophagy Process

Some studies clarified the role of ERLINs within ER lipid-raft domains in different cell functions. In this regard, since ERLINs are exclusively protein markers of ER lipid raft-like microdomains [78] and MAMs represent an ER sub-compartment containing lipid rafts involved in both stress and metabolic signaling [61,98], it was of interest to understand the role of ERLINs in regulating MAMs activity.

The localization on the ER membrane of ERLINs was characterized by their fractionation in cholesterol-enriched, detergent-resistant membrane [78] through a membrane domain at their N-terminus, which is also responsible of their binding to the ER lumen in 1000-kD hetero-multimeric complexes [80,88,91], as well as to MAM level, as recently reported [72]. This specific inter-organelle localization of ERLIN1 at MAM level has been demonstrated to be governed by mitofusin 2 (MFN2), a membrane tethering bridge, acting at microdomains [99].

Moreover, our previously published data revealed that the embedding of MFN2 in “raft-like microdomains” could be an essential event in the mitochondrial network extension induced by Mdivi-1 [41].

Recently, a crucial role for ER-mitochondria association in autophagy initiation was also suggested. Autophagy is a highly dynamic and well conserved lysosome-dependent mechanism of degradation, which is distinct from other degradative processes, where components are degraded to provide internal source of nutrients for energy [100,101]. During autophagy process, an isolation membrane sequesters a small portion of the cytoplasm, long-lived proteins and superfluous or excess organelles, to form the autophagosome. Finally, autophagosome fuse with lysosomes to yield autolysosomes, which degrade internalized materials by resident hydrolases.

In recent years, the sources of autophagosome membrane have sparked a great interest, although the membrane donors for autophagosomes biogenesis need even more investigations. In fact, since autophagy is an unselective bulk degradation process, the specific membrane origin of all autophagosome remains unclear, although morphological features of autophagosomes are fundamentally common to conventional and alternative autophagy. The plasma membrane (PM), mitochondria, ER-mitochondria contact sites, ER-Golgi intermediate compartment (ERGIC), Golgi apparatus, and endoplasmic reticulum exit sites (ERES), have been suggested to provide lipids to the growing isolation membrane in mammalian cells, but the exact mechanism underling this process remains obscure. Recently, Manganelli et al. [72] identified the presence of raft-like microdomains in the MAMs, which could be crucial in the mitochondria-ER interplay, leading to autophagosome formation. Furthermore, they found a molecular association of the ganglioside GD3, already considered to be a paradigmatic “brick” of lipid rafts, with two core-initiator proteins of autophagy, thus autophagy and beclin1 regulator 1 (AMBRA1) and beclin1 (BECN1) respectively [65]. AMBRA1 is a WD40-containing protein playing a prominent role in the development of the central nervous system. AMBRA1 binds to BECN1 and stabilizes BECN1/Vps34 complex, thus potentiating its lipid kinase activity and promoting the formation of autophagosome [100]. The association with BECN1 does not occur in the WD40 domain of AMBRA1 but resides in a central region of the protein (aa 533–780), which is also enough to induce autophagy when overexpressed [102]. Indeed, AMBRA1 shows a dynamic interaction with the dynein motor complex during autophagy induction [103]. AMBRA1 is linked, together with BECLIN1 and Vps34, in an autophagy inactive state to the dynein complex via a specific association with dynein light chains (DLCs) 1 and 2, which is mediated by two DLC-binding consensus motifs (TQT) at the C-terminal region of AMBRA1. Upon autophagy induction, AMBRA1 is dissociated from the dynein complex upon ULK1-dependent phosphorylation to translocate on ER membrane together with Beclin1-Vps34 leading to autophagosome biogenesis. A physical interaction between ERLIN1 and AMBRA1 after autophagy triggering was demonstrated by coimmunoprecipitation and FRET analysis [72]. In addition, a correlation was reported between the domains of AMBRA1 that are able to mediate the association with ERLIN1, i.e., the AMBRA1 central region and AMBRA1 C terminal and the ability of these protein fragments to promote autophagy. Interestingly, depletion of gangliosides significantly hindered both AMBRA1-ERLIN1 interaction and nutrient deprivation induced autophagy, suggesting that combination of ERLIN1 and lipids within MAMs may regulate recruitment and activity of distinct sets of proteins involved in the autophagic process (Figure 2) [72]. It can be hypothesized that, under autophagic stimulation, a recruitment of these molecules into lipid rafts at the MAM level could take place. This could represent a prerequisite for membrane scrambling between mitochondria and ER, thus suggesting a function for MAMs in the earliest events leading to omegasome formation [65]. Thus, interaction of ERLIN1 with AMBRA1 within MAM leads to consider this protein as a new interesting player in the autophagy machinery. Moreover, the localization of ERLINs within ER lipid rafts might prompt reflection on further considerations, since several components of lipid rafts are also involved in intracellular homeostasis of Ca^2+^ of which the ER is the main store.

The multifunctional organelle ER maintains Ca^2+^ homeostasis, which is necessary for suitable functioning, including lipid and protein biosynthesis, protein folding, posttranslational modification, and regulation of gene expression [104]. Studies have shown that several Ca^2+^ dependent pathways are involved in autophagy triggering. Indeed, numerous Ca^2+^ origins involve various downstream effectors, containing protein kinase C, Ca^2+^/calmodulin-dependent kinase β (CaMKKβ or CaMKK2), ERK, and Vps34 (a calmodulin protein) [105,106]. Thus, the proposed role of ERLINs in the degradation of the calcium channel (inositol 1,4,5-triphosphate receptor) could therefore explain their role in the autophagy mechanism.

On the other hand, ERLIN2 is part of the ERAD pathway, which is responsible for the degradation of misfolded proteins in the ER [107]. Autophagic elimination and ERAD occur independently and exert protective roles, promoting cell survival. Thus, autophagy and ERAD, in concert, contribute to eliminate toxic species of misfolded and accumulated proteins from the ER [108,109]. In particular, loss of ERLIN2 function is therefore expected to result in accumulation of aberrant proteins. Suppression of ERLIN2 by RNA interference markedly inhibited IP_3_ receptor polyubiquitination and degradation and the processing of other ERAD substrates.

Overall, these studies have identified ERLINs as a key ERAD pathway component and suggested that it may act as a substrate recognition factor. Conceivably, ERLINs could play a role in ERAD by recruiting ERAD pathway components to membrane microdomains, thereby facilitating the assembly and spatial regulation of the multiprotein complexes that mediate ERAD.

## 5. ERLINs: New Actors in the Crosstalk between Autophagy and Apoptosis in the Cancer

The role of autophagy in removing damaged proteins and organelles is considered as an essential tool to limit their cumulative harmful effects inside cells. Therefore, it is not surprising that autophagy defects characterize many human tumors [110,111,112]. On the other hand, stress-activated autophagy may favor survival of tumor cells, mostly when apoptosis is defective. In the light of these considerations, ERLINs could play a crucial role in the interplay between autophagy and apoptosis. Indeed, ERLIN1 was suggested to regulate tumor progression in the fibrosarcoma cell line (2FTGH). On this regard, silencing of ERLIN1 has been associated to a significant reduction of the number of both autophagosomes and autolysosomes under starvation, indicating a decreased autophagy flux. Moreover, ERLIN1 has been shown to affect the autophagy-dependent survival under chemotherapy treatment; whereby, in ERLIN1-silenced cells the treatment with cisplatin induced an increase of apoptosis as verified by PARP cleavage and propidium iodide staining [72].

Along similar lines, previous studies also confirmed that the depletion of nutrients and growth signals, a condition likely associated with oncogenesis and ER stress, results in increased ERLIN2 production in breast epithelial cells. Wang and colleagues [113] found that amplification of the ERLIN2 gene and its resulting overexpression occurs in both luminal and Her2^+^ subtypes of breast cancer, suggesting a role for ERLIN2 as a novel oncogenic factor under the ER stress response pathway.

The ER has evolved highly specific signaling pathways, collectively termed the “unfolded protein response” (UPR), to ensure protein folding fidelity and to protect the cell from ER stress. During cellular stress conditions, including nutrient deprivation and dysregulation of protein synthesis, cells accumulate unfolded/misfolded proteins within the ER lumen, leading to activation of the unfolded protein response (UPR). The UPR signaling cascade relies on three major stress sensors located onto ER membrane namely RNA-dependent protein kinase-like kinase (PERK), activating transcription factor 6 (ATF6), and inositol-requiring enzyme 1α (IRE1α). Therefore, adapting to ER stress through translational attenuation, upregulation of ER chaperones, and protein degradation take place [114,115,116]. Depending on the type or degree of the stress, cells activate different UPR pathways to realize a survival or death fate [117]. In addition, cancer cells may adapt to the cellular stress and evade stress-induced apoptotic signaling by differentially activating the UPR branches. As part of the UPR program, ER-associated degradation (ERAD) targets aberrantly folded proteins in the ER. Recent studies provide evidence that UPR and ERAD components are highly expressed in various tumors, including human breast cancer [118,119,120]. During tumor development and progression, increased amounts of misfolded proteins caused by gene mutations, hypoxia, nutrient starvation, and high levels of reactive oxygen species which inevitably lead to ER stress [119,120,121,122]. The activation of UPR and ERAD induces an adaptive response exploited by tumor cells as a mechanism of resistance to ER stress.

Notably, the tumor microenvironment has been shown to cause a steady level of ER stress response in cancer cells, ultimately, promoting tumor development and metastasis [123]. Multiple stressors within the tumor microenvironment can cause ER stress in tumor cells. They include both intrinsic tumor attributes, such as hypoxia, oxidative stress, and nutrient deprivation and external stressors, such as chemotherapy, radiation, and immunotherapy. Cancer cells then utilize effective pathways to respond, adapt, and save themselves from ER stress-induced cell death.

Moreover, recent studies revealed that ER stress could also impede the efficacy of anti-cancer treatment including immunotherapy by manipulating the tumor microenvironment. Thus, an increasing number of chemotherapy resistance mechanisms involved in ER stress have been discovered. ER stress-related molecular markers, such as GRP78, PERK, IRE1α, have been reported to have prognostic values for cancer patients [124].

Since aberrant UPR and ER stress are major contributors to cancer development, chemoresistance, and poor prognosis, there has been strong interest in clinically influencing this process as a strategy to restrain tumor growth and reverse drug resistance. Overall, generation of high-quality small molecules to modulate ER stress or to target the UPR, either as a monotherapy or in combination with chemotherapy, targeted therapy, and immunotherapy have shown promising preclinical treatment efficacy and offering the opportunity to develop therapeutics for a vast range of ER stress-related diseases [124,125].

Finally, recent evidence has indicated that overexpression of ERLIN2 is modulated by the IRE1α/XBP1 axis in the ER stress pathway favoring the adaptation of breast epithelial cells to this condition by supporting cell growth. Thus, ERLIN2 could facilitate a cytoprotective response to various cellular stresses associated with oncogenesis, although the molecular mechanisms by which ERLIN2 coordinates ER pathways in breast carcinogenesis remain unclear.

## 6. ERLINs in Neurodegenerative Diseases

Emerging studies on mutations in the ERLIN1 or ERLIN2 genes have related these variants to rare neurodegenerative diseases, e.g., hereditary spastic paraplegia (HSP) [126,127,128], a heterogeneous group of genetic neurodegenerative disorders (MND) characterized by progressive spasticity that primarily affects the lower extremities and afterwards it can extend cranially with progressive spasticity of the lower limbs [129]. In addition, a dominant form of pure HSP was lately related to a heterozygous mutation in ERLIN2, which underlines the complexity of the ERLIN2 mutation in spastic paraplegia (SP) phenotype [128].

On this regard, Alazami et al. identified a novel ERLIN2 nullimorphic deletion that defines the SPG18 locus, a particular form of HSP, characterized as a juvenile primary lateral sclerosis [130]. In fact, the authors revealed that the loss of the ERLIN2 initiation exons along with mislocalization of exon 2 was sufficient to cause a nullimorphic allele. As a consequence, the loss of ERLIN2 function as a key component of the ERAD control promoted a persistent activation of IP_3_R and neuronal channels due to an impairment of proteasomal ubiquitination of IP_3_Rs and degradation. This is not surprising since this mechanism was already demonstrated by a recruitment of ubiquitin E3 ligase RNF170 which constitutively binds to the ERLIN1/2 complex [131]. In this way, since a well-defined function of the two proteins/complex is to mediate IP_3_R processing, its deficiency might lead to increased IP_3_-dependent signaling via IP_3_Rs, followed by increased and potentially prolonged Ca^2+^-release from the ER. Thus, the chronic perturbation of IP_3_R levels and Ca^2+^ handling could contribute to neurodegeneration [132]. In fact, Ca^2+^ signaling is now emerging as a common pathophysiological pathway in several studies, to which it has been attributed a role for therapeutic target in a broad range of neurodegenerative diseases, including Alzheimer’s disease [133], Huntington’s disease [134], and spinocerebellar ataxias (SCA) [135].

In according with these findings, other neurological disorders of the upper motor neuron with onset in early childhood, including “intellectual disability”, motor dysfunction and joint contractures (IDMDC)” were also related to a deficiency expression of ERLIN2 [130,136], although it cannot be ruled out that ERLIN2 may exert its pathogenic role through the impaired degradation of other proteins; in fact, this mechanism remains to be investigated in ALS phenotype [137].

Next, further studies have better clarified that although both ERLIN1 and ERLIN2 can assemble to bind RNF170, only ERLN2 contains the motif able to interact with IP_3_R, since cells expressing ERLIN2 mutations on Thr-65 region (T651) strongly reduced the PI3P-binding capacity of the ERLIN1/2 complex with a deleterious effect on cell health, leading to a growth suppression of cultured motor neurons as evident in the hereditary spastic paraplegia disease [89].

## 7. ERLINs in Viral Infections

Since the ERLIN proteins are associated with ER raft-like microdomains, they may function as ER-anchoring factors. Interestingly, Inoue and Tsai reported a link between ERLIN1/2 and viral infections. The non-enveloped polyomavirus SV40 penetrates the ER membrane to reach the cytosol and causes infection. Upon entry, SV40 dynamically recruits B12, an ER transmembrane J-protein, as well as other membrane components, into discrete puncta in the ER membrane called foci that correspond to viral cytosol entry sites. In this study the authors demonstrated that this traffic is regulated by ERLINs. In fact, ERLIN1 and ERLIN2 (ERLIN1/2) are cellular components that bind to B12 and facilitate B12’s reorganization into the SV40-induced foci. Indeed, when ERLIN1/2 were silenced, they found B12 into the nucleus suggesting that ERLINs could act as anchors, restricting B12 in the ER [138].

Yet, a recent article demonstrated a role for ERLIN1 in hepatitis C virus (HCV) infection. Indeed, ERLIN1 is considered a host factor required for HCV life cycle; depletion of ERLIN1 leads to a decreased infection efficiency. Since the dependence of both HCV on lipid metabolism and the ER for its life cycle, ERLIN1 could regulate not only early steps leading to RNA and protein accumulation, but also later steps affecting virus production [139].

## 8. Conclusions

At present, the role of ERLINs in cellular functions is poorly clarified. However, recent advances have indicated that the ERLIN1/2 complex regulates calcium channeling, cell cycle progression, and cholesterol homeostasis of the cell, playing a role in the regulation of viral infections.

In this way, this complex may be considered a new player in the control of cell fate by protecting the cells from ER-induced cell death and regulating the autophagy process, by interaction with key molecules of the autophagosome machinery, including AMBRA1, within lipid rafts at the MAM level.

Further studies are needed to investigate the interplay between lipids and ERLINs in regulating membrane organization and function.

A better understanding of the molecular mechanisms by which ERLINS coordinate ER pathway(s) and their possible role in disease(s) pathogenesis might provide an important opportunity to identify new therapeutic strategies for the treatment of several disease (s).

## Figures and Tables

**Figure 1 cells-10-02408-f001:**
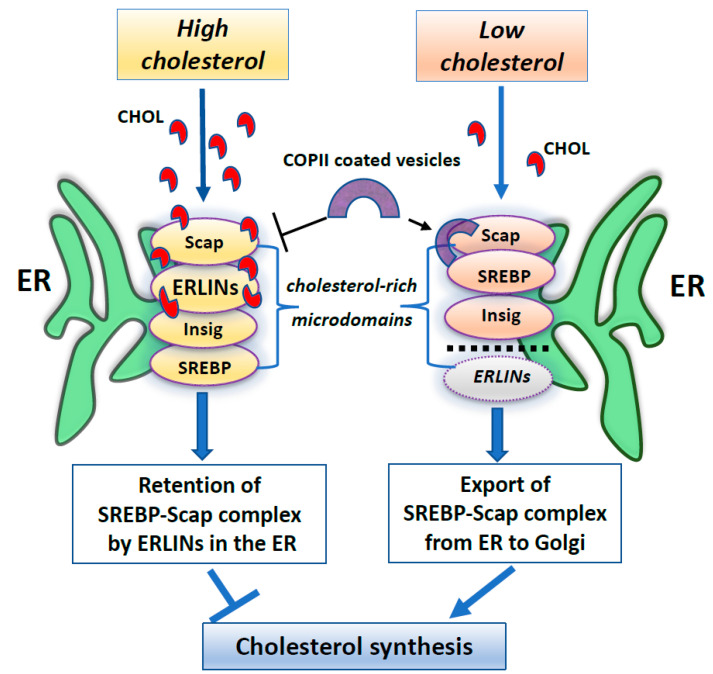
A summary scheme showing ERLINs as novel regulators of SREBP machinery. High levels of cholesterol promote the stability of the SREBP–Scap–Insig complex by ERLINs-cholesterol binding at ER cholesterol-rich microdomains. When cholesterol is depleted, COPII proteins coat clusters of SREBP-Scap complexes excluding ERLINs and facilitating their vesicular transport to the Golgi.

**Figure 2 cells-10-02408-f002:**
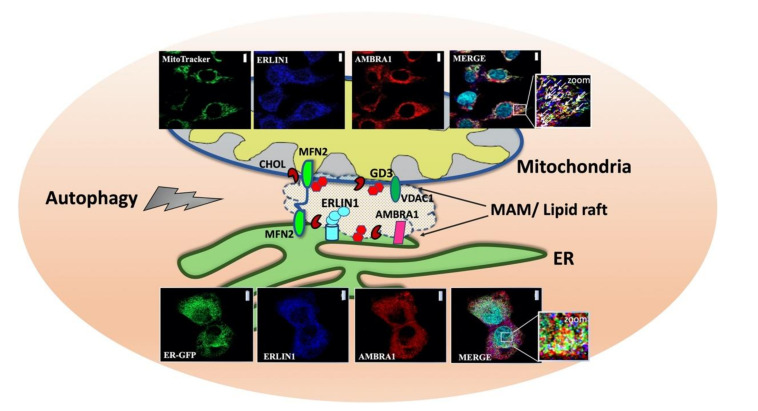
Association of ERLIN1 with raft-like microdomains upon amino acid starvation (HBSS)-induced autophagy. Schematic drawing depicting raft-like microdomains within ER-mitochondria contact sites (MAMs). Confocal microscopy images showing increased colocalization between AMBRA1 (red), ERLIN1 (blue) and a specific mitochondrial tracker (green) or an ER marker (ER-GFP) (green) respectively upon autophagy induction. Nuclei were stained with DAPI (Turquoise). To note, overlapping areas resulting from green, red and blue fluorescence in merge micrograph (see arrows in magnification of the boxed areas) indicate that colocalization of ERLIN1-AMBRA1-MitoTracker or ERLIN1-AMBRA1-ER-GFP respectively increases in cells treated with HBSS as compared to untreated cells. Images were acquired using a LSM 900, Airyscan SR Zeiss confocal microscopy and the co-localization was measured using the ZEN 3.0 Blue edition software and expressed as µm^2^ per cell. Scale bar: 10 µm. Confocal images are reproduced with permission from REF.72 TAYLOR & FRANCIS. Abbreviations: ERLIN1, ER lipid raft-associated protein 1; AMBRA1, autophagy and beclin1 regulator1, MFN2, mitofusin 2, GD3, ganglioside; CHOL, cholesterol, VDAC1, voltage-dependent anion channel 1.

## Data Availability

Not applicable.

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
