# Peer review of "Role of ERLINs in the Control of Cell Fate through Lipid Rafts"

_cells, 2021, doi:10.3390/cells10092408_

Round 1
Reviewer 1 Report
The authors in this review highlighted the importance of ERLINs proteins in the control of cell fate and their peculiar localization within mitochondria-associated endoplasmic reticulum membranes. These regions are characterized by the presence of raft-like subdomains with a specific role in membrane scrambling and function. Despite the interesting topic, several major revisions are required before acceptance.
The second chapter of this review, in my opinion, needs to be re-organized since, in the current form, lacks fluidity and clearness. I suggest to divide the chapter in several subchapters, each shortly analyzing the raft-like micro-domains of a specific organelle. For example, the chapter can be divided as follows:
2.1 Rafts-like microdomains in the mitochondria
2.2 Rafts-like microdomains in the nuclei
2.3 Rafts-like microdomains in Golgi and ER
2.4 MAMs raft-like microdomain
Few published manuscripts reporting the importance and role of lipid microdomains in other organelles should also be mentioned in this chapter:
PMID: 15030566
PMID: 29427074
PMID: 33330457
PMID: 33715815
There are few mistakes I spotted in the references (please double check all of them again):
Line 201 in the sentence: ``stomatin binds cholesterol`` reference 81 should be removed since this work is about stomatin-like proteins 2 binding to cardiolipin.
Reference 89 title is wrong: the protein is SPFH2 not PFH2.
Line 219/226. The authors write that several studies showed that ERLIN-2 is the dominant partner in the ERLIN 1/2 complex and contains the determinants for binding to activated IP3Rs and PI3P.However, only one reference is reported at the end of the paragraph. Moreover, authors should modify this paragraph because there is too much overlapping text with the cited manuscript (Wright et al.).
In paragraph 227-246 the authors describe the interaction between SREBP and other proteins involved in its regulation. This is a crucial pathway in the regulation of cholesterol homeostasis, therefore I think a figure showing the localization and interaction of each protein described in the paragraph, in condition of low cholesterol and high cholesterol, could be very beneficial for the readers.
In chapter 4, the authors in figure 1 showed confocal pictures of co-localization during starvation between ERLIN1 and MFN2. However, these experimental data are not related to reference 97 as reported in the text and I failed to connect them to any published manuscript. Figure legend is basically the description of the methodology. If the data are published permission of the reproduction should be reported (like done in figure 2). If these are unpublished data, figure must be removed. In any case, since this is a review, I think is more appropriate to replace this figure with a different one like the above mentioned picture explaining the SREBP interaction pathway.
In chapter five, the author should elaborate more on the role of ER stress in influencing the tumor microenvironment and potential consequence of it (including the reduction in efficacy of anti-cancer treatment). Consequently ER stress and UPR could be effective therapeutic targets. Please see:
PMID: 31320759
PMID: 34136492
One aspect completely not mentioned by authors is the emerging role of ERLIN proteins in the regulation of viral infection (PMID: 31810281 PMID: 28614383)
Maybe this is a topic that can be discussed in the conclusion session
Reviewer 2 Report
Manganelli et al. wrote a timely and stimulating overview of the role of ERLINs in organization of ER’s and MAM’s lipid raft-like domains and their role in autophagy, apoptosis and neurodegenerative disease. The literature review is nearly comprehensive, and many interesting possibilities are discussed.
While the main part of the manuscript devoted to ERLINS is well written and provides specific and useful information, a more general overview of intracellular lipid raft-like domains in section 2 is rather vague, providing little specifics and it is limited to general statements. The reader will benefit from a more concise but more to the point, short introduction to internal membrane organization.
It would be helpful to offer the reader in Conclusions a list of unresolved questions and future directions in ERLINs research.
Minor:
Please spell out SPFH.
Not every reader will make a connection between cell starvation and HBSS treatment. Please explain.
In Fig. 2 legend, “the co-localization between ERLIN1, AMBRA1 and Mito-344 Tracker was measured using the ZEN 3.0 Blue edition software and expressed as mm2 per cell” seems to belong to a deleted part of the figure.
Several typos: “It is note that,” “Underling,” “anti-anti-ERLIN1 (blue) and AMBRA1 340 (red) antibodies."
Round 2
Reviewer 1 Report
The authors performed the requested changes (including figure 1 showing the role of ERLINs in regulating the SREBP pathway). The quality of the manuscript improved.